

# Experimental assessment of the relationship between rainfall intensity and sinkholes caused by damaged sewer pipes

Tae-Young Kwak[1], Sang-Inn Woo[2], Choong-Ki Chung[3], and Joonyoung Kim[4]

[1]Seismic Safety Research Center, Korea Institute of Civil Engineering and Building Technology, Goyang-si, Gyeonggi-do 10223, South Korea.
[2]Department of Architectural & Civil Engineering, Hannam University, Daedeok-gu, Daejeon 34430, South Korea.
[3]Department of Civil & Environmental Engineering, Seoul National University, Gwanak-gu, Seoul 08826, South Korea.
[4]Division of Smart Interdisciplinary Engineering, Hannam University, Daedeok-gu, Daejeon 34430, South Korea.

*Correspondence to*: Joonyoung Kim (goldenrain91@gmail.com)

## ABSTRACT

In several countries, the rising occurrence of sinkholes has led to severe social and economic damage. Based on the mechanism of sinkhole development, researchers have investigated the correlation between rainfall intensity and sinkholes caused by damaged sewer pipes. In this study, the effect of rainfall intensity on the formation of eroded zones, as well as the occurrence of sinkholes caused by soil erosion due to groundwater infiltration through pipe defects, has been analyzed through model tests. The ground in Seoul was adopted using weathered granite soil, which is generally used for backfill sewer pipes, and groundwater levels corresponding to three different rainfall intensity conditions were considered. The ground level changes and ground displacements were measured continuously, and the particle image velocimetry (PIV) algorithm was applied to measure the displacement at each position of the model ground. The results indicate that impeding the excessive rise of groundwater levels by securing sufficient sewage treatment facilities can effectively prevent the development of sinkholes caused by pipe defects.



## 1 Introduction

In recent times, cases of sinkholes (or ground cave-ins) have been reported in several countries, such as the US, Japan, Italy, South Africa, China, and South Korea. Major social and economic issues have ensued owing to the resulting structural problems, such as the collapse of buildings and road erosion (Bae et al., 2016; Galloway et al., 1999; Gao et al., 2013; Guarino and Nisio, 2012; Intrieri et al., 2015; Kuwano et al., 2010a; Oosthuizen and Richardson, 2011; Yokota et al., 2012). In general, sinkholes can be classified into two types: (1) natural sinkholes and (2) anthropogenic sinkholes. Natural sinkholes occur when the underlying ground layer (e.g., karst landscape) is easily soluble in water, whereas anthropogenic sinkholes occur in a non-karst environment, caused by human activity such as sewage damage, inadvertent excavation, or groundwater lowering.

Both types of sinkholes have similar mechanisms, and the detailed process of occurrence is as follows (Brinkmann et al., 2008; Caramanna et al., 2008; Kuwano et al., 2010a; Martinotti et al., 2017; Oosthuizen and Richardson, 2011; Rogers, 1986): (1) A cavity is formed underground by external factors (the water-soluble ground layer dissolves in groundwater to cause a natural sinkhole, or soil erosion occurs along with the groundwater outflow due to sewage damage or excavation to cause an anthropogenic sinkhole). (2) The groundwater level rises during rainfall and falls after the rainfall, causing the soil around the cavity to be lost and the cavity to expand. (3) A sinkhole is finally generated because of the repeated increase and decrease of the groundwater level.

Based on the mechanism for both types of sinkholes (natural and anthropogenic), a direct relationship can be inferred between the rainfall intensity, which leads to the transition of the groundwater level (rise and fall), and the occurrence of sinkhole. Notably, the change in climate due to global warming has resulted in higher rainfall intensity with fewer rainy days (Alpert et al., 2002; Kristo et al., 2017; Rahardjo et al., 2019a, 2019b). In South Korea, the maximum daily rainfall has increased over the decades in most regions and is expected to increase significantly in the future (Choi et al., 2017; Nadarajah and Choi, 2007; Wi et al., 2016). Thus, there is a growing need to study the correlation between rainfall intensity and sinkhole occurrence.

Martinotti et al. (2017) showed that a period of torrential rain and the rainfall intensity triggered the natural sinkholes in Italy. Gao et al. (2013) confirmed that the groundwater level rise due to extremely heavy rainfall has a significant effect on sinkhole generation in a karst environment in China. Van Den Eeckhaut et al. (2007) showed that the formation of numerous natural sinkholes in Belgium corresponded with periods of high rainfall and high groundwater recharge, which commonly increased the weight of the overburden and decreased its cohesion.

The majority of sinkholes in non-karst environments are known to occur because of damaged sewer pipes. In Seoul, South Korea, an average of 677 sinkholes and subsidence occurred annually from 2010 to 2015, of which 81.4 % were due to damage to old sewer pipes (Bae et al., 2016). In Japan, local governments in sewage projects were surveyed to identify cases of subsidence due to damage to sewer pipes. As a result, a total of 17,000 data were reported from 2006 to 2009 (Yokota et al., 2012).



Considering these factors, several researchers have conducted statistical analysis and model experiments to investigate the correlation between rainfall intensity and sinkholes caused by damaged sewer pipes. Kwak et al. (2016) showed that the number of anthropogenic ground cave-in cases increased with the increase in total monthly precipitation. In addition, it was confirmed that ground cave-ins are prone to occur after exceptionally heavy rains. By quantifying Pearson's correlation coefficient between two relevant observations, Choi et al. (2017) showed that the monthly accumulated precipitation and the quantity of subsidence are related to a certain extent. Guo et al. (2013) and Tang et al. (2017) used model experiments and evaluated the effect of the defect size, groundwater level, and particle size on soil erosion due to groundwater infiltration through pipe defects. However, they only used non-cohesive soils and covered extreme cases with groundwater levels significantly exceeding the ground level.

In this study, the urban area in Seoul has been simulated, and model tests have been conducted to analyze the effect of rainfall intensity on the formation of eroded zones, as well as the occurrence of sinkholes caused by soil erosion due to groundwater infiltration through pipe defects. The model ground was constructed using weathered granite soil (which is generally cohesive), which is mainly used to backfill the sewer pipes in Seoul. Three rainfall intensity conditions (heavy rainfall, very heavy rainfall, and extremely heavy rainfall) were set for the groundwater level, based on summer rainfall patterns in Korea, to be applied in the model tests. The groundwater level change, discharged soil volume, and ground displacement were measured continuously throughout the tests. In particular, the particle image velocimetry (PIV) method, which can continuously measure and analyze the displacements in the ground, was applied to quantify the ground deformation with the occurrence and expansion of underground cavities.

The remainder of this paper is organized as follows. Sect. 2 describes the model test device, model grounds, and test conditions. Sect. 3 discusses the model test results. Finally, Sect. 4 presents the conclusions and contributions of this paper.

## 2 Experimental program

### 2.1 Experiment apparatus

In this study, experiments were performed using the model tester developed by Kwak et al. (2019) to simulate ground subsidence (Figure 1). The distance between each pipe in the sewer pipe network in Seoul was examined and found to be around 1.2 m. In order to exclude the effects of unnecessary boundary conditions, the width of the model soil was set to 1.4 m, with respective left and right margins of 0.1 m. Considering that the average landfill depth of a sewage pipe in Seoul is 0.9 m (Kim et al., 2018), the soil chamber was built to a height of 1.0 m, including a 0.1 m clearance to facilitate sample composition. The depth of the soil chamber was set to 0.1 m to simulate the plane strain condition, and the front plate of the chamber was made of acrylic plate to allow the inside of the ground to be photographed during the test.





A slit was installed at the bottom of the soil chamber to simulate the damage of the sewer pipe allowing the
inflow and outflow of sewage and the outflow of soil during the model test. The width of the slit was set to
2 cm, based on the study by Mukunoki et al. (2012), such that $B/D_{max}$ was 4.2 (the maximum particle diameter
of weathered soil $D_{max} = 4.76$ mm). A supply valve and a drain valve were installed under the slit to control
the inflow and outflow of groundwater as well as the outflow of eroded soil. The external water tank
connected to the inlet valve was designed to maintain a constant level even when water is continuously
supplied to the model ground. Through experimental assessment (Table 1), the National Diaster Management
Institute of Korea (2014) suggested a relationship between the rainfall intensity and the hydraulic head in the
sewage network conditions near Gangnam station. It should be noted that the hydraulic head increases
linearly with the rainfall strength until the rainfall strength is 40 mm/h, but thereafter increases sharply. In
the present study, the height of the external tank was made adjustable to simulate the various rainfall intensity
(related to hydraulic head).

## 112     2.2 Model ground

The vast majority of prior studies that have experimentally assessed the ground subsidence and sinkhole due
to sewer pipe damage have been conducted on poorly-graded non-cohesive soils (Guo et al., 2013; Indiketiya
et al., 2017; Kuwano et al., 2010a, 2010b; Sato and Kuwano, 2015; Tang et al., 2017). However, in several
countries, the sewage reclamation specifications allow the landfill soil to contain 15–25 % of fine contents.
There are no restrictions on particle size distribution apart from the maximum particle size and #No. 4 sieve
passing (Japan Road Association, 1990; Ministry of Environment of Korea, 2010). In the present study, to
simulate the ground in Seoul in which weathered granite soil, which is a well-graded cohesive soil, is widely
distributed, the model ground was created by collecting Gwanak weathered soil and adjusting the fine content
to 7.5% to meet the fine content standard. The degree of compaction was also set to 93 % of the standard
maximum unit dry weight $\gamma_{d,max}$ to satisfy the sewer pipe landfill standards, and the model ground was
constructed with the optimum moisture content. Figure 2 shows the particle size distributions of the adjusted
and natural Gwanak soil in comparison with the sewer pipe landfill standards in South Korea and Japan.
Table 2 lists the basic physical properties, strength parameters and saturated permeability coefficient of the
adjusted Gwanak soil used in the model test.

## 127     2.3 Digital image analysis

In geotechnical engineering, digital imaging techniques are primarily used to measure the deformation of
target samples (Alshibli and Sture, 1999; Indiketiya et al., 2017; Kim et al., 2017; Kwak et al., 2019; White
et al., 2003). In the present study, the displacement at each position of the model ground was measured by
applying the PIV algorithm (Adrian, 1991), which is the most widely used technique in the field of
geotechnical engineering. The PIV cross-correlation on the pixel sets of the pre-deformation and post-
deformation images were calculated to obtain the point with the highest correlation. The position of the





sample set with the highest correlation is used to estimate the relative displacement at each position of the
sample (Kim et al., 2011; White et al., 2003). In this study, the internal displacement of the sample was
evaluated using GeoPIV (White and Take, 2002), a commercial program that is widely used to apply the PIV
technique in geotechnical engineering. With the displacement, the volume and shear strain are estimated
together for the analysis.
In general, when applying the PIV technique, high accuracy analysis results can be obtained when the
uniqueness of the pixel set increases as the size of the pixel set increases. However, in order to calculate
displacements at various positions, it is necessary to set an appropriate size for the set of pixels. Accuracy
and precision verification of the GeoPIV program was performed for this, and the size of the pixel set was
set to 100 by 100 pixels as a result. As shown in Figure 3, the PIV technique was applied to the positions of
a total of 2600 pixel subsets (65 by 40). To minimize the boundary effect between the interface of the sample
and the soil chamber, the vicinity of the wall was excluded from the analysis. In addition, any excessive
relative displacement due to soil erosion (no highly correlated pixel sets found in the post-deformation image)
was excluded from the analysis.
**2.4 Test procedures**
Once the model ground was created, the model tests which consisted of multiple cycles were conducted.
Each experimental cycle consisted of a water supply stage that simulated the rainfall and a water drainage
stage that simulated the aftereffect of the rainfall. During the water supply stage, water from the external
water tank was introduced into the soil chamber through the supply valve to reach the target groundwater
level. Following this, during the water drainage stage, the supply valve was closed and the drainage valve
was opened to allow the soil to discharge through the lower slit along with the water. Table 3 shows the
conditions of the three model tests conducted in this study, simulating cases with rainfall intensities of 40
mm/h and 50 mm/h presented in Table 1, as well as that with the groundwater level rising to the ground
surface.
Linear variable displacement transducers (LVDTs) were installed at three locations on the surface of the
ground, at 0, 30, and 60 cm from the center of the soil chamber, to measure the surface displacement during
the tests (Figure 1). During the model tests, digital images of the ground were continuously captured, and the
PIV technique was applied to analyze the displacement and deformation (Adrian, 1991; Alshibli and Akbas,
2007; Kim et al., 2017; Kwak et al., 2019). In addition, the amount of soil discharged through the slit was
measured after the water supply and water drainage stages of each test.



### 3 Experimental results and discussion

### 3.1 Test 1: Heavy rainfall intensity (47 cm hydraulic head)

#### 3.1.1 Water supply stage

Test 1 was conducted by introducing groundwater to a 47 cm initial hydraulic head (the height difference between the slit and weir in the water tank) to simulate a heavy rainfall intensity of 40 mm/h. In the water supply stage of Test 1, no soil deformation occurred on the ground surface (measured by the LVDTs) and in the ground (measured by the PIV technique) as the groundwater level approached 47 cm. Immediately after opening the slit, the water pressure acting on the ground directly above the slit was 4.5 kPa, and the vertical earth pressure generated by the upper soil was about 16.7 kPa. Therefore, under this condition, the soil always had a positive effective stress, and the piping phenomenon did not occur. In this study, since the model ground was densely constructed ($D_R$ = 78 %) with a sufficient degree of compaction ($R_C$ = 93 %) according to domestic specification, no water compaction (Kwak et al., 2019), which occurs mainly when sewage flows into a loose sandy soil, was observed. From these results, it was confirmed in this experimental case that the resistance factor (due to the soil strength parameter) was greater than the sum of the drag force (upward force by infiltration pressure during water supply) and the gravity (downward force).

#### 3.1.2 Water drainage stage

In the water drainage stage of Test 1, no soil deformation was observed on the ground surface as in the water supply stage. The deformation in the ground was evaluated by applying PIV to the images captured during the test. Figures 4, 5, 6 are the PIV analysis results showing the estimated displacement vector, volume, and shear strain increments in six phases: (a) 0–30 s, (b) 30–60 s, (c) 60–90 s, (d) 90–120 s, (e) 120–150 s, and (f) 150–180 s. For the volumetric strain, the red grid (the area with positive values) indicates that the area has expanded, and the blue grid (the area with negative values) indicates that the area has been compressed.

In the water drainage stage, the water pressure applied through the slit disappeared, and the groundwater in the soil chamber was discharged quickly through the slit. Unlike in the water supply stage, the ground below the groundwater level became saturated and lost its apparent cohesion. The rapid outflow of groundwater resulted in a downward infiltration into the ground, and the soil was discharged from the area immediately above the slit, where there was no active restraining pressure (and thus, no shear strength), along with the groundwater.

During the initial phase of the water drainage stage (0–60 s), the soil was discharged through the slit, causing a downward displacement in the periphery of the cavity, and a triangular cavity was formed just above the slit (Figure 4 (a) and (b)). In addition, volume and shear strain increments occurred intensively around the cavity (Figure 5 (a), (b), Figure 6 (a) and (b)). In the 60–90 s interval of the water drainage stage, as shown in Figure 4 (c), the soils on the both sides of the cavity collapsed, and the cavity expanded laterally. The





volume and shear strain increments were concentrated in small areas near the cavity, similar to the initial
stage (Figure 5 (c) and Figure 6 (c)).
As shown in Figure 4 (d), during the 90–120 s interval of the groundwater drainage stage, the lateral
expansion of the cavity inside the ground was completed, and no downward displacement was observed in
the upper part of the cavity and the soils on the sides. The volume and shear strain increments were also not
observed in the outer region of the cavity (Figure 5 (d) and Figure 6 (d)). In this phase, the cavity collapsed;
the soil accumulated near the slit gradually shifted to escape into the slit, and the deformation was
concentrated near the slit.  After 120 s, the soil regained its apparent adhesion due to surface tension, and its
outflow stabilized as the drainage completed. Finally, a mushroom-shaped cavity was formed (Figure 4 (e)
and (f)).
**3.2 Test 2: Very heavy rainfall intensity (70 cm hydraulic head)**
**3.2.1 Water supply stage**
Test 2 was conducted by setting the maximum groundwater level to 70 cm to simulate a high rainfall intensity
of 50 mm/h. During the water supply stage of Test 2, no soil deformation was observed on the ground surface
and in the ground by both LVDT and PIV analyses. As a result, owing to the soil strength parameter, the
resistance factor was found to remain greater than the sum of the drag force (upward force by infiltration
pressure during water supply) and the gravity (downward force), despite the application of a higher hydraulic
pressure in Test 2 as compared to that in Test 1.
**3.2.2 Water drainage stage**
During the water drainage stage of Test 2, no vertical displacement was observed on the surface of the model
ground. The displacement of the soil element according to the development of the underground cavity was
observed by the PIV technique. Figures 7, 8, and 9 show the displacement increment vectors, incremental
volumetric strain distribution, and incremental shear strain distribution, respectively; the analysis was
conducted in four phases: 0–30 s, (b) 30–60 s, (c) 60–90 s, and (d) 90–120 s (the displacement ended within
120 s).
In the initial phase of the water drainage stage (0–30 s), the soil was discharged through the slit, causing an
internal collapse near the slit. Thus, an underground cavity was formed (Figure 7 (a)), differing from that in
Test 1 in terms of shape as well as location; it was located close to the maximum groundwater level (about
60 cm from the bottom plate). These results indicate that the hydraulic pressure (related to rainfall intensity)
affects the shape and location of the underground cavity in the water drainage stage. In Test 1, the eroded
zone was formed up to about 89 % of the maximum groundwater level. In Test 2, it developed up to about
86 %. When a poorly-graded non-cohesive soil was used under the same experimental conditions, the cavity
developed up to 107 % of the maximum groundwater level (Kwak et al., 2019). This shows that the well-
graded cohesive soil used in this study has a greater resistance to soil erosion. In addition, during the initial



stage (0–30 s), the incremental volumetric and shear strains were found to be concentrated in the upper area
of the underground cavity (Figure 8 (a) and Figure 9 (a)).
During the 30–90 s phase, downward displacement was no longer observed at the top of the cavity;
displacement in the slit direction occurred only in the left and right areas adjacent to the cavity (Figure 7 (b)
and (c)). The volumetric and shear strains also showed a tendency to be concentrated in the left and right
areas where the displacement occurred, indicating that the cavity gradually increased laterally (Figure 8 (b),
(c), Figure 9 (b), and (c)). In the process of forming a cavity, the downward infiltration pressure was low,
and the soil that had lost strength accumulated near the slit. On the other hand, when the downward infiltration
pressure was higher, all the soil that had lost strength escaped, resulting in the formation of an oval cavity.
After 90 s, as the groundwater level was exhausted, the unsaturated strength of the ground was restored, and
no further displacement or deformation were observed inside the ground (Figure 7 (d), Figure 8 (d), and
Figure 9 (d)).
**3.3 Test 3: Extremely heavy rainfall intensity (90 cm hydraulic head)**
**3.3.1 Water supply stage**
Test 3 was conducted to simulate the intensity of an extremely heavy rainfall that causes the groundwater
level to rise up to the surface of the ground. In the water supply stage of Test 3, significant displacements
were measured on the surface (LVDTs) and inside the model ground (PIV). Figure 10 shows the surface
displacement over time, with a gradual subsidence after approximately 2400 s. The ground displacements
identified by the PIV technique from 0–2000 s also showed no specific behaviors. Therefore, the internal
displacement vectors identified as a result of the PIV technique after 2000 s are shown in Figure 11, overlaid
onto the final photograph of each step: (a) 2000–2400 s, (b) 2400–2800 s, (c) 2800–3200 s, and (d) 3200–
3600 s.
As the groundwater level reached about 75 cm (83 % of ground height), soil particle displacement was
observed in the soil from 2000–2400 s. This result indicates that, owing to the strength of the soil, the
resistance factor becomes smaller as the model ground is saturated, and the weight of the soil in the saturated
region cannot be supported. Since the soil in the upper part of the groundwater level still maintained its
unsaturated strength, the downward displacement appeared only in the area adjacent to the groundwater level.
There was still no subsidence observed on the surface (Figure 11 (a)).
From 2400–2800 s, downward displacement towards the slit was observed throughout the soil area. In
particular, a larger downward displacement was observed in the inverted triangle region above the slit, which
was significantly affected by the inflow of groundwater (Figure 11 (b)). As the groundwater level rose, the
matric suction expressed in the unsaturated region of the ground decreased. Therefore, the subsidence on the
ground surface was also measured from this phase. From 2800–3200 s, the groundwater level reached 80 cm
from the bottom (89 % of ground height), and the maximum downward displacement of the entire water
supply stage was observed during this phase (Figure 11 (c)). This indicates that infiltration occurs when the



groundwater level approaches the ground surface, and the soil structure is no longer supported as there is no
longer sufficient matric suction in the ground directly above the groundwater level. After 3200 s, downward
displacement occurred continuously throughout the soil area until groundwater level reaches the target level
(Figure 11 (d)).

### 3.3.2 Water drainage stage

The water drainage stage of Test 3 was divided into four phases for the analysis: (a) 0–30 s, (b) 30–60 s, (c)
60–90 s, and (d) 90–120 s. The displacement increment vectors, incremental volumetric strain distributions,
and incremental shear strain distributions of each stage are shown in Figure 12, 13 and 14, respectively,
overlaid onto the photograph of the target ground taken at the end of each phase.
In the initial phase (0–30 s) of the water drainage stage of Test 3, the groundwater was rapidly discharged
into the slit owing to high downward infiltration pressure. As the soil particles escaped along with the
groundwater discharge, the upper ground collapsed, forming an anthropogenic sinkhole similar in shape to
the punching shear failure (Figure 12 (a)). In the previous tests, the cavities formed up to about 86 % and
89 % of the maximum groundwater level. However, in this case, the upper soil layer became inordinately
thin and eventually collapsed inwards. The shape of the formed anthropogenic sinkhole indicated significant
downward displacement (of the soil that had lost strength) towards the slit. The sudden collapse of the ground
clogged the slit, which in turn prevented soil discharge. At this time, the shear deformation also showed a
tendency to be concentrated around the collapsed soil (Figure 14 (a)). After the soil was completely drained,
no significant deformation inside the ground and on the ground surface were observed via the PIV technique
and the LVDTs after 30 s, as the matric suction allowed the ground to recover its unsaturated strength.

### 3.4 Comparative Study

To quantitatively analyze the effect of rainfall intensity on ground cavity and sinkhole development, the
evolution of the cavity size with time in the water drainage stage was obtained for each test, and the time at
which the water was completely drained was also displayed, as shown in Figure 15. For the hydraulic pressure
of 45 cm and 70 cm, the time taken for the groundwater to drain completely was 70 s and 90 s, respectively.
However, in Test 3, although the groundwater level was higher, the soil collapsed instantly, resulting in an
anthropogenic sinkhole, and the time taken for complete drainage was 80 s, which was faster than that in Test
2. After the drainage was completed, the cavity sizes measured in Test 1 and Test 2 were 497 cm$^2$ (66 % of
the final cavity size of 742 cm$^2$) and 1286 cm$^2$ (87 % of the final cavity size of 1482 cm$^2$), respectively. In
both Tests 1 and 2, the cavity expanded for about 30 s after the drainage was completed, at which time its
size tended to stabilize. In Test 3, where the anthropogenic sinkhole occurred, a cavity of 1207 cm$^2$ (56 % of
the final cavity size of 2171 cm$^2$) was formed after the drainage was completed, after which the cavity
continued to expand for approximately 200 s.




Table 4 shows the ratio of the weight of the total soil volume to the weight of the discharged soil volume, the
volume ratio of the area corresponding to the cavity, and the weight ratio of the loosening zone, respectively.
The size and internal density change of the loosening zone were calculated by the following method. (1)
After completion of the test, the discharged soil was dried to measure the weight. (2) The weight of the area
corresponding to the cavity was calculated by multiplying the calculated volume of the cavity by the initial
density of the soil. The soil weight corresponding to the loosening zone was calculated through the difference
between the results of steps (1) and (2).  (3) The size of the loosening zone was calculated by excluding the
area corresponding to the cavity from the area overlapping with the volumetric strain calculated in each step.
(4) The internal density change was confirmed using the results of steps (2) and (3).
As shown in Table 4, the size and density change of the loosening area were found to be nearly identical in
the three tests. On the other hand, as the hydraulic head increased, the weight and volume of the eroded zone
and the average width of the cavity relative to the slit width increased linearly. However, recalling the fact
that the hydraulic head increased drastically when the rainfall intensity exceeds a certain threshold, it can be
inferred that the volume of the discharged soil and the size of the eroded zone may also increase exponentially
with rainfall intensity. The threshold value is definitely specific to a given sewer-system. Thus the
experimental results of this study suggest that to prevent sinkholes caused by pipe defects, sewage pipe
network facilities need to be expanded to inhibit the rapid rise of groundwater levels in preparation for
increased torrential rain caused by climate change.
**4 Conclusions**
In this study, model tests were used to analyze the effects of rainfall intensity on the formation of the eroded
zone and the occurrence of sinkholes caused by soil erosions due to groundwater infiltration through pipe
defects. The model tests were conducted to simulate the actual site conditions as far as possible by using the
soil used around sewer pipe networks and the sewer pipe landfill standards as well as a large-scale soil
chamber. The groundwater level was applied to the model tests by setting three hydraulic heads based on the
heavy rainfall characteristics of South Korea: (1) heavy rainfall intensity (47 cm hydraulic head); (2) very
heavy rainfall intensity (70 cm hydraulic head); and (3) extremely heavy rainfall intensity (90 cm hydraulic
head). Throughout the model tests, the groundwater level changes and the ground surface displacements were
measured continuously from the start to the end of the tests. In addition, the PIV technique, which can
continuously measure and analyze the displacement of the entire ground, was applied to quantify the ground
deformation (volumetric strain and shear strain), generation, and expansion of the underground cavity. Based
on the results of the three tests, the following observations were drawn:
(1) The rainfall intensity considerably affected on the ground deformation during and after a rainfall.
(2) Under heavy and very heavy rainfall intensity conditions, no internal soil deformation occurred while the
groundwater level was rising. However, under extremely heavy rainfall intensity conditions, ground
subsidence was observed. This result indicates that the resistance factor (due to the soil strength parameter)





334 becomes smaller than the sum of the drag force (upward force by infiltration pressure during water supply)

335 and the gravity (downward force) when the rainfall intensity exceeds a certain threshold, which was found

336 to have a hydraulic head between 70 cm and 90 cm under the given system.

337 (3) After heavy rainfall (that leads to the rise of the groundwater level due to the infiltration of groundwater

338 through the sewer pipe defects), the soil was discharged from the area above the slit with the rapid outflow

339 of groundwater, where there was no active restraining pressure. During the formation and development of

340 cavity along with the drop in the groundwater level, the incremental volumetric and shear strains were

341 concentrated in the vicinity of the underground cavity.

342 (4) The height and average width of cavities increased linearly with the applied hydraulic head, and notably,

343 sinkhole opened under extremely heavy rainfall intensity. Referring the previous study which showed the

344 relationship between the hydraulic head and rainfall intensity, the discharged soil and the size of the eroded

345 zone may increase exponentially with rainfall intensity.

346 It should be noted that the hydraulic head-rainfall intensity relationship used in this study is site-specific. The

347 induced hydraulic head under the same rainfall intensity can be different site to site. Nevertheless, the

348 experimental observations of this study confirm the influence of rainfall intensity on the soil erosion near the

349 sewer pipe defects as well as sinkhole occurrence and suggest a necessity of sewage pipe network facilities

350 rehabilitation in preparation for increased torrential rain caused by climate change.

**Author contribution**

352 The conceptualization was done by TYK, CKC, and JK planned methodology. TYK performed the analysis

353 using software, and validation was performed by SIW and CKC. JK performed formal analysis. TYK

354 prepared the original draft, while all authors contributed to the review and editing. Visualization and graphics

355 were designed by TYK and JK. SIW and CKC supervised the research work.

**Competing interests**

357 The authors declare that they have no conflict of interest

**Acknowledgements**

359 This research was supported by the Research Institute at the college of Engineering of Seoul National

360 University. In addition, the support of Jin-Tae Han, research fellow of the Korea Institute of Civil Engineering

361 & Building Technology, is greatly appreciated.



**Financial support**
This research was supported by a grant (code: 20SCIP-C151438-02) from Construction Technologies
Program funded by Ministry of Land, Infrastructure and Transport of Korean government. Also, this work
was supported by the National Research Foundation of Korea (NRF) grant funded by the South Korean
government (MSIP) (No. 2015R1A2A1A01007980).

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





**Figure 1: Schematic of the model test device.**


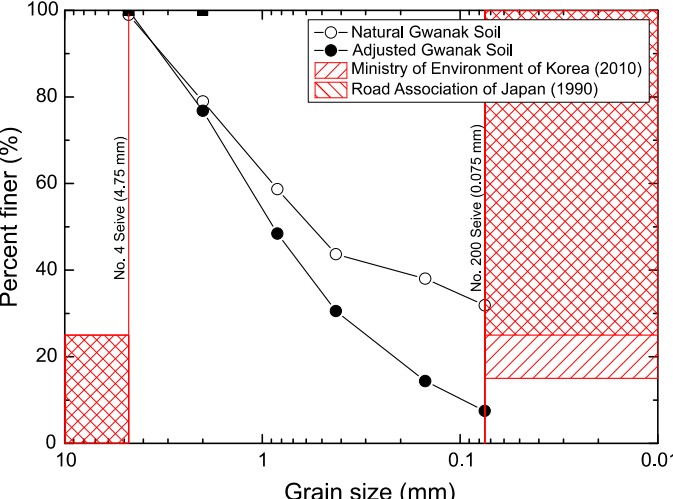


**Figure 2: Grain distribution of the natural Gwanak soil and the Gwanak soil adjusted as per the requirements**
**for backfill materials in South Korea (Ministry of Environment of Korea, 2010) and Japan (Japan Road**
**Association, 1990).**




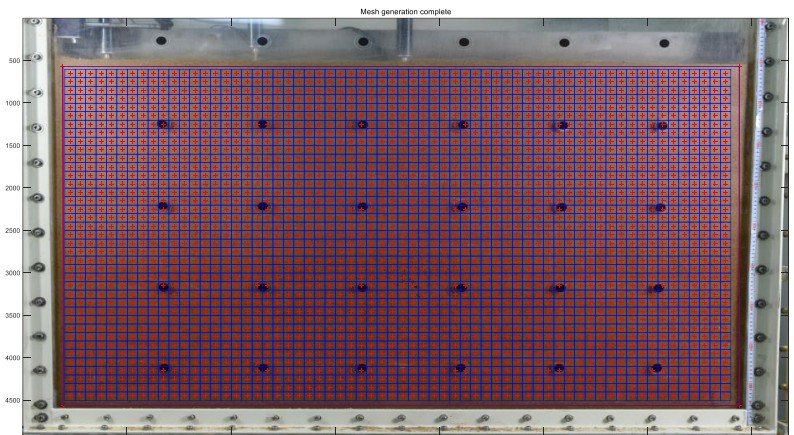


**Figure 3: Selected pixel subsets and center points for digital image analysis.**


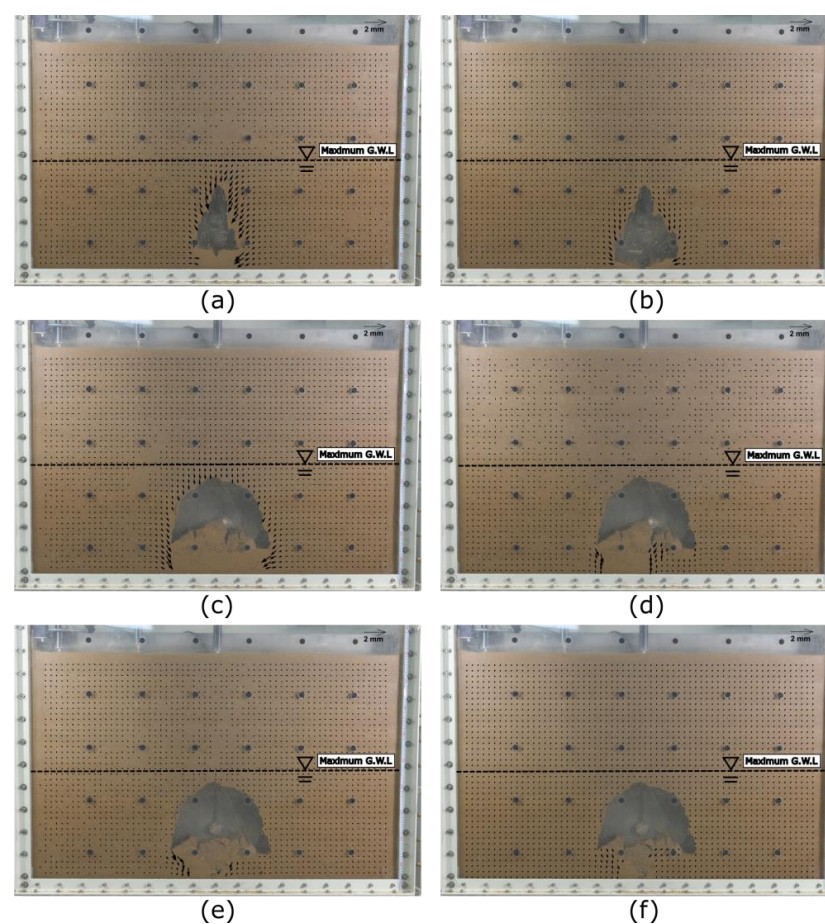


**Figure 4: Displacement increment vectors inside the model ground for Test 1 during the water drainage stage:**
**(a) 0–30 s, (b) 30–60 s, (c) 60–90 s, (d) 90–120 s, (e) 120–150 s, and (f) 150–180 s.**




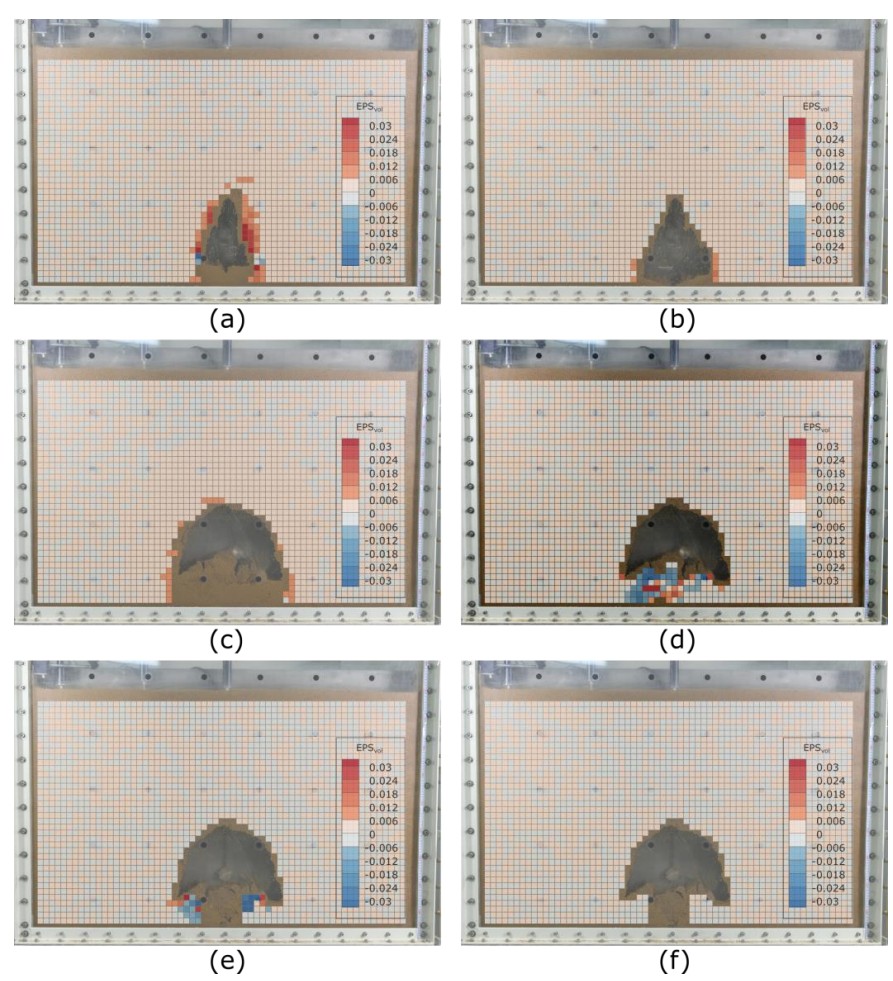


**Figure 5: Volumetric strain inside the model ground for Test 1 during the water drainage stage: (a) 0–30 s (b) 30–60 s, (c) 60–90 s, (d) 90–120 s, (e) 120–150 s, and (f) 150–180 s.**

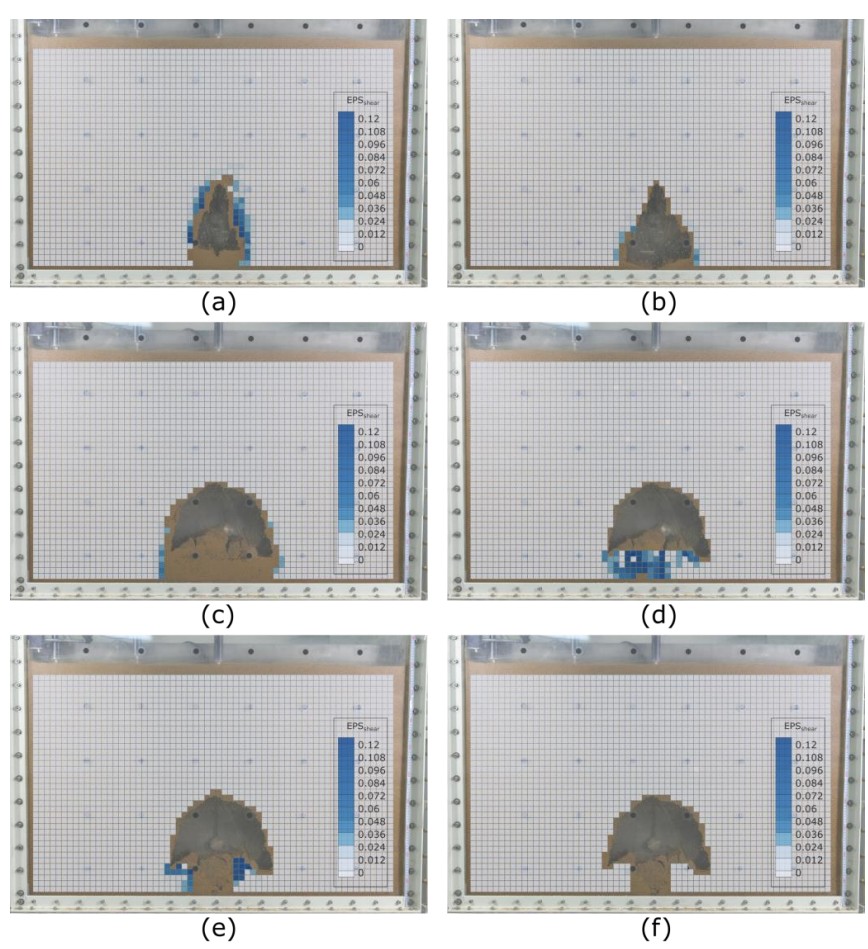

**Figure 6: Shear strain inside the model ground for Test 1 during the water drainage stage: (a) 0–30 s, (b) 30–60 s, (c) 60–90 s, (d) 90–120 s, (e) 120–150 s, and (f) 150–180 s.**

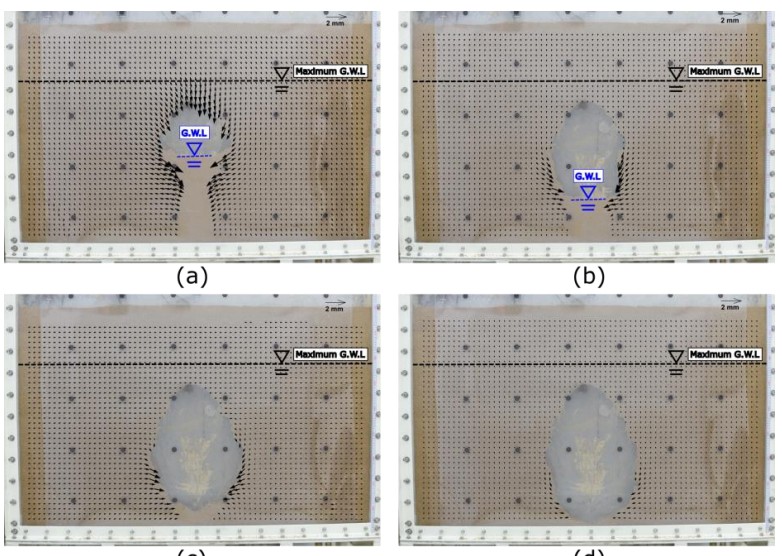

**Figure 7: Displacement increment vectors inside the model ground for Test 2 during the water drainage stage:**
**(a) 0–30 s, (b) 30–60 s, (c) 60–90 s, and (d) 90–120 s.**

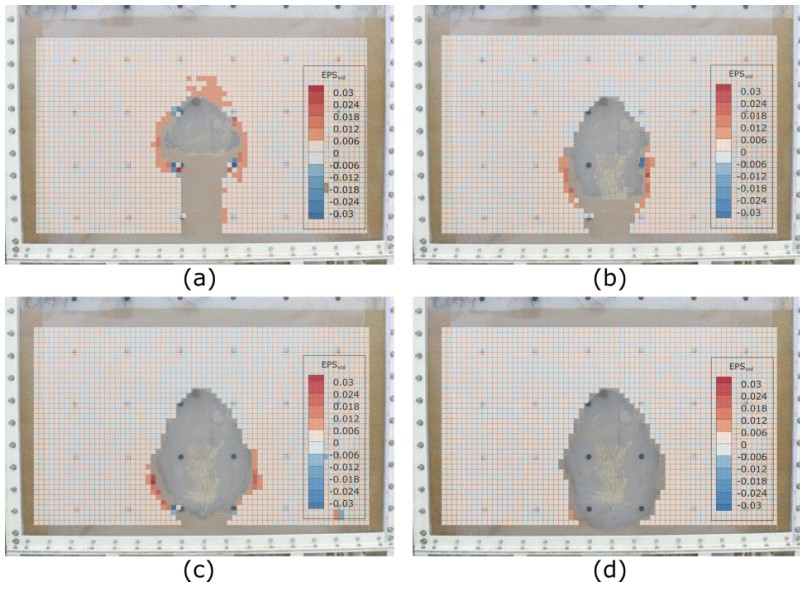

**Figure 8: Volumetric strain inside the model ground for Test 2 during the water drainage stage: (a) 0–30 s, (b)**
**30–60 s, (c) 60–90 s, and (d) 90–120 s.**


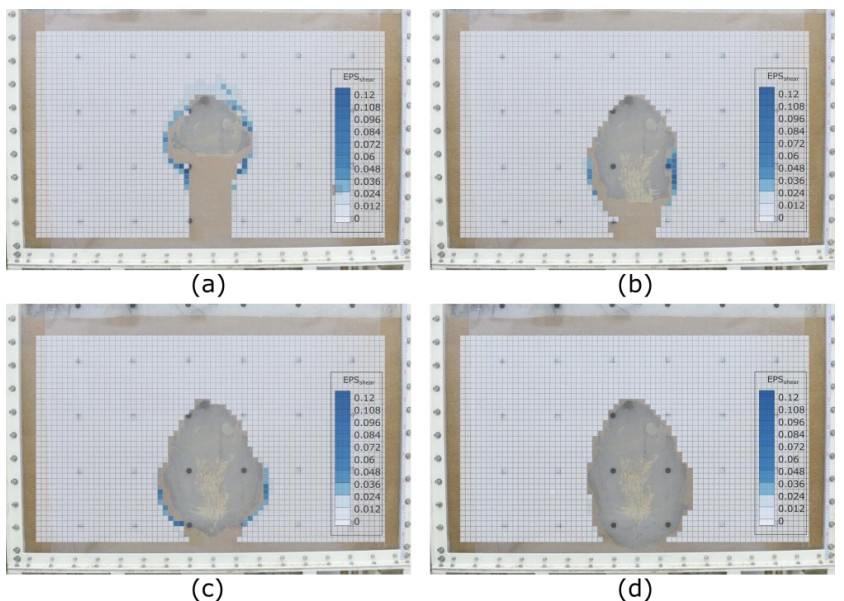


**Figure 9: Shear strain inside the model ground for Test 2 during the water drainage stage: (a) 0–30 s, (b) 30–60 s, (c) 60–90 s, and (d) 90–120 s.**


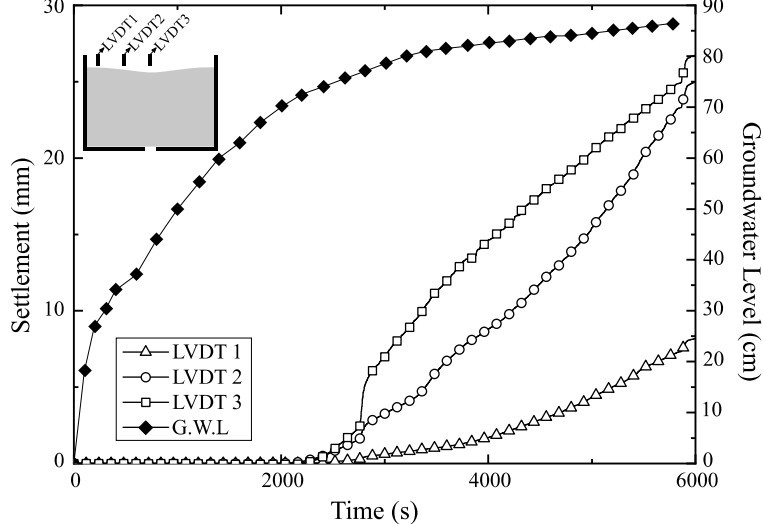


**Figure 10:  LVDT measurement during the water supply stage (Test 3).**




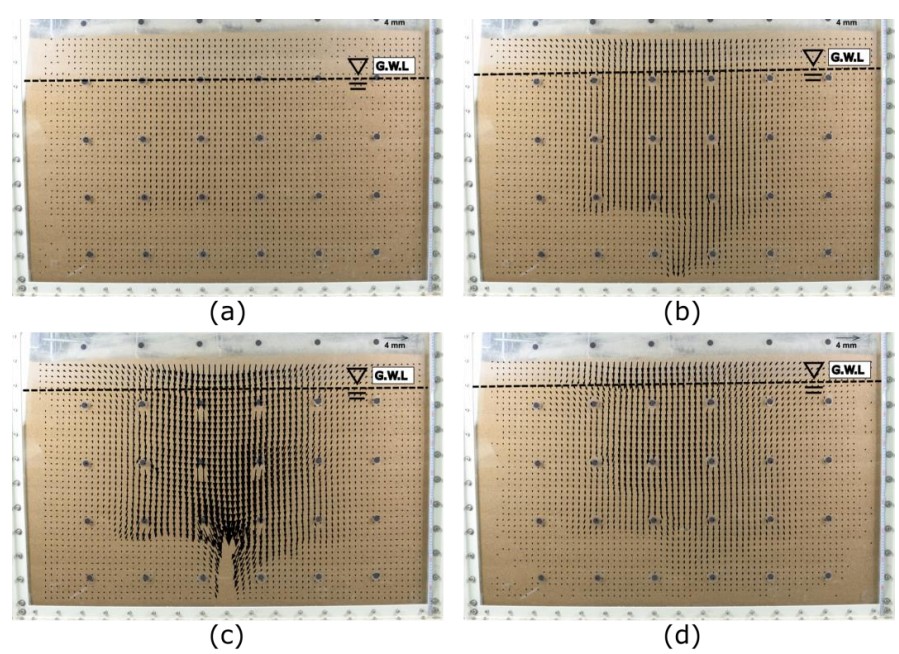


**Figure 11: Displacement increment vectors inside the model ground for Test 3 during the water supply stage: (a) 2000–2400 s, (b) 2400–2800 s, (c) 2800–3200 s, and (d) 3200–3600 s.**


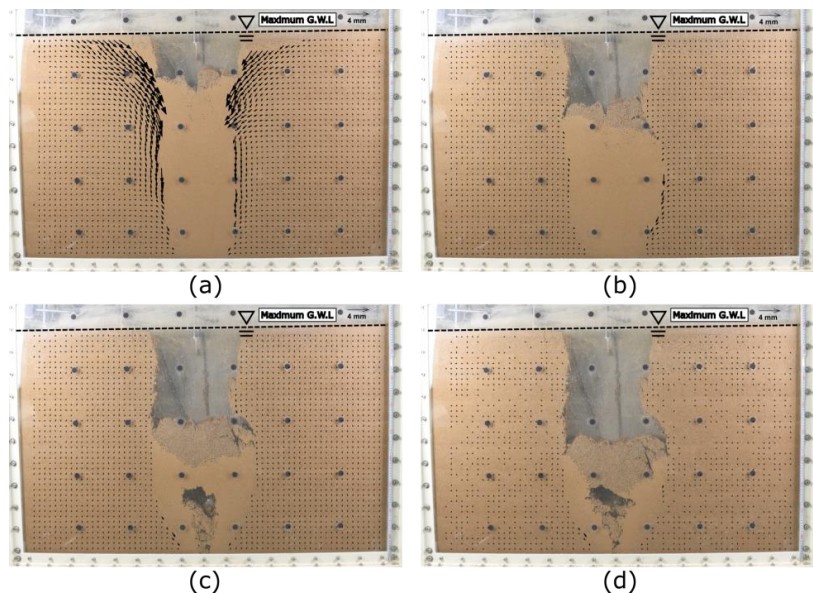

**Figure 12: Displacement increment vectors inside the model ground for Test 3 during the water drainage stage: (a) 0–30 s, (b) 30–60 s, (c) 60–90 s, and (d) 90–120 s.**

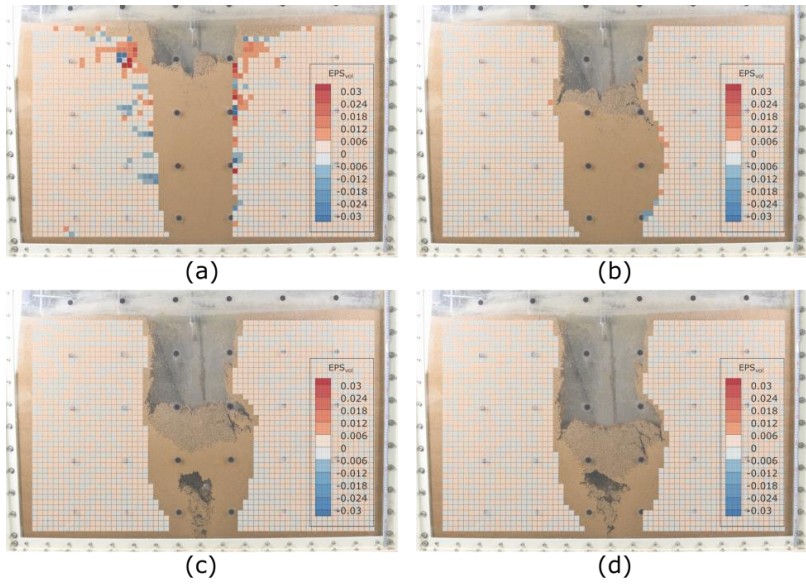

**Figure 13: Volumetric strain inside the model ground for Test 3 during the water drainage stage: (a) 0–30 s, (b) 30–60 s, (c) 60–90 s, and (d) 90–120 s.**




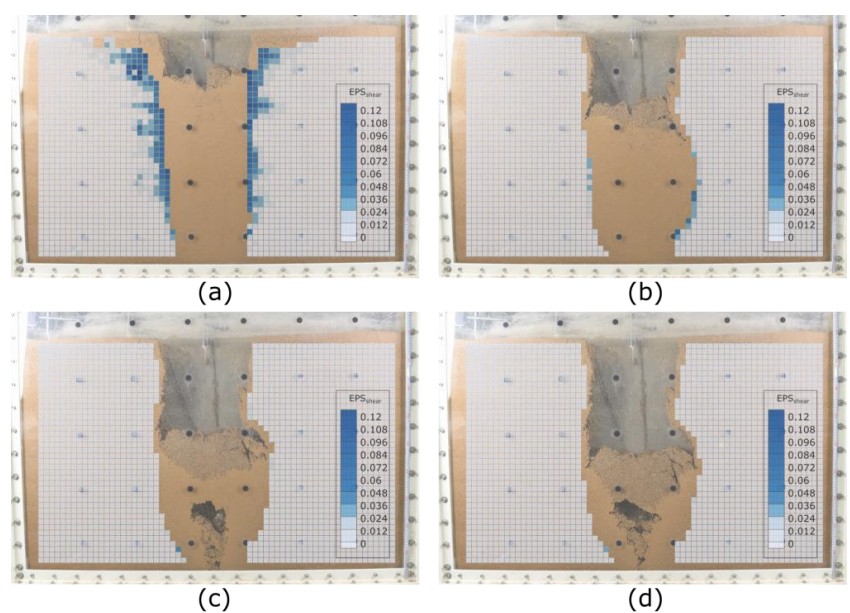


**Figure 14: Shear strain inside the model ground for Test 3 during the water drainage stage: (a) 0–30 s, (b) 30–60 s, (c) 60–90 s, and (d) 90–120 s.**



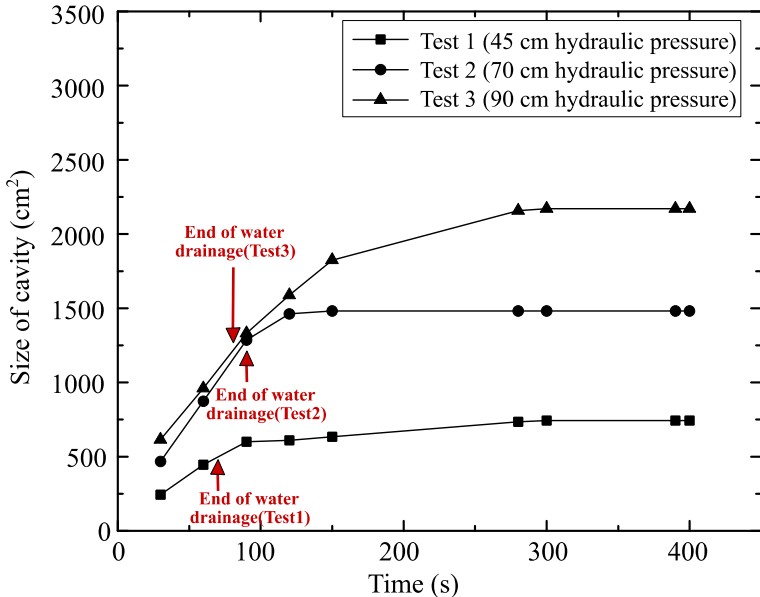


**Figure 15: Sizes of cavities developed during the water drainage stage in each test.**




**Table 1: Relation between rainfall intensity and hydraulic head applied to sewer pipes**
**(National Diaster Management Institute of Korea, 2014).**

| Rainfall intensity | Hydraulic head |
|---|---|
| 20 mm/h | 33 cm |
| 30 mm/h | 40 cm |
| 40 mm/h | 47 cm |
| 50 mm/h | 70 cm |


**Table 2: Properties of adjusted Gwanak soil.**

| Description | | Adjusted Gwanak soil (Fine content 7.5 %) |
|---|---|---|
| Classification in USCS (Unified Soil Classification System) | | SW-SM |
| Specific gravity $G_S$ | | 2.62 |
| Mean grain size $D_{50}$ (mm) | | 1.013 |
| Coefficient of curvature $C_C$ | | 1.24 |
| Coefficient of uniformity $C_U$ | | 12.4 |
| Standard maximum dry unit weight* $\gamma_{d,max}$ (kN/m³) | | 18.5 |
| $e_{max}$ / $e_{min}$ | | 0.96 / 0.39 |
| Void ratio | | 0.51 |
| Optimum water content* (%) | | 11.4 |
| Strength parameter** | Saturation S 100 % | Cohesion $c$ (kPa) | 3.9 |
| | | Friction angle $\phi$ (°) | 36.3 |
| | Saturation S 44.2 % | Cohesion $c$ (kPa) | 15.8 |
| | | Friction angle $\phi$ (°) | 38.3 |
| Saturated permeability coefficient $k_{sat}$ (cm/s) | | $1.45 \times 10^{-4}$ |

* Estimated from the standard compaction tests
** Estimated from the direct shear tests; S = 44.2 % corresponds to $w_{opt}$ obtained from the standard compaction tests



**Table 3: Model test conditions used in this study.**

| Test No. | Soil type | Slit size | Degree of compaction $D_C$ (Relative density $D_R$) | Burial depth | Maximum groundwater level |
|---|---|---|---|---|---|
| #1 | | | | | 47 cm |
| #2 | Adjusted Gwanak soil | 2 cm | 93 % (78 %) | 90 cm | 70 cm |
| #3 | | | | | 90 cm |

**Table 4: Comparative studies of the model tests.**

| Test | Test 1 (47 cm G.W.L) | Test 2 (70 cm G.W.L) | Test 3 (90 cm G.W.L) |
|---|---|---|---|
| Percentage of the weight of the discharged soil in the total initial weight of the model ground | 6.4 % | 12.9 % | 18.3 % |
| Percentage of the volume of the eroded zone(cavity or ground cave-in) in the total initial volume of the model ground | 5.9 % | 12.5 % | 18.2 % |
| Ratio of average cavity width to slit width | 11.5 (22.9 / 2 cm) | 13.1 (26.2 / 2 cm) | 16.4 (32.8 / 2 cm) |
| Average density change in the loosening zone | −3.1 kN/m3 | −3.7 kN/m3 | −2.9 kN/m3 |