# Peer review of "Experimental assessment of the relationship between"

_Natural Hazards and Earth System Sciences, 2020_

## Short Comment (SC1) · 25 Jul 2020

1. General comments This paper reports a series of experiments to analyze the sinkhole formation associated with rainfall intensity by simulating a leakage of an underground damaged sewer pipe. A slit at the bottom of the experiment chamber was considered as damage of pipe, and three different rainfall intensities were designed by controlling the hydraulic head connected to the slit of the chamber. The ground settlement was measured, and the deformation of soils around the pipe is captured by the particle image velocimetry (PIV) technique. Overall, the authors in this work present a rising issue of the sinkhole and its relationship to rainfall by utilizing an experimental

model set-up. I believe this paper will be of interest to the audience and would support publication after the following comments are addressed.

1) Authors should clearly address and explain how the test procedure is designed to simulate the rainfall and the sewer pipes. a. In the test procedure, the hydraulic head was selected as a variable to represent the rainfall intensity, which eventually formed different target groundwater levels. Therefore, the amount of water introduced into the chamber and the duration of water supply stage may indicate additional information related to the rainfall intensity. For example, if the water supply stage of Test 3 took longer than that of Test 1, this set-up may not properly reflect the actual rainfall intensity and its influence on the groundwater level around the pipes. In addition, the flow from the damaged sewer pipe may not be the only source of water supply into the underground. b. In this work, the damage of sewer pipes is approximated as a slit of which the size is determined followed by the previous study. While this damage could be the main source of water supply into the ground, the drainage may not occur through this damage. In other words, the water drainage set-up using the slit and drainage valve may lead to an extra discharge of soils and water. If the groundwater is discharged through a thin slit at the bottom of the chamber, it may be easily expected that the soil around the slit may easily collapse and deformed.

2. Specific comments 1) Line 149: What's the meaning of multiple cycles? 2) Line 173: As point out by the authors, the test-set up is likely to the piping simulation. Then, is it possible to analyze the sinkholes and rainfall intensity though the piping analysis? For example, using the critical hydraulic gradient? 3) Line 255: How is the resistance factor determined? If there's an equation, it could be helpful for readers. 4) Line 262: Is the matric suction analyze in a quantitative manner? Because of cohesion, this may limitedly affect the behavior of soils. 5) Lines 279-280: The meaning is not clear.

---

## Referee Comment (RC1) · Anonymous Referee #1 · 29 Jul 2020

**1. General comments to authors**

This paper examined the effect of rainfall intensity on the sewer-related soil erosion and its evolution by means of model tests and image analysis. In order to reflect the field conditions in South Korea, the backfill material, rainfall intensity, and compaction criteria were considered in the model tests. The topic is clear and suitable with the subject of this journal. There are, however, several aspects that need to be improved, especially in relation with the test procedure and actual sewer-related soil erosion. Revising the manuscript considering the following comments are also recommended.

**2. Specific comments to authors**

[Figure]

Q1: The terms 'ground cave-in' and 'sinkhole' are used interchangeably which are recommended to be unified.

Q2: L101"The width of the slit was set to 2 cm, based on the study by Mukunoki et al. (2012), such that B/Dmax was 4.2." Justify the width of the slit the authors determined in relation with the listed reference.

Q3: L107-108. Some clarification on the condition of "Relationship between the rainfall intensity and the hydraulic head in the sewage network conditions near Gangnam station" are needed. I wonder if this condition has been sufficiently considered in the model tests of this study.

Q4: L117. typo (#No. 4 sieve passing).

Q5: L141-142. Additional information on the validation of PIV technique, such as accuracy, analysis condition, will be of interest to the readers.

Q6: L149. Explanation about the multiple cycles is required. I believe that one cycle consisted of water supply and drainage stage and it was repeated, but the manuscript contains only the result of one cycle.

Q7: Additional information on the rainfall record which can prove the suggested three rainfall intensities in this study are realistic will enhance the credit of this paper.

Q8: The procedure of calculating the average cavity width in Table 4 is not clear.

---

## Referee Comment (RC2) · Anonymous Referee #2 · 6 Aug 2020

An experimental study about the relationships between rainfall intensity and development of sinkholes caused by damaged sewer pipes in Korea is described in the paper. The topic is certainly of interest to NHESS, and the work contains interesting data and considerations. I have listed in the accompanying file a number of small corrections, and a few requests of clarification on some issues that are not clear to me.

I discourage throughout the manuscript the use of the term "ground cave-ins", since this is not used in the international literature, and may induce confusion and misunderstandings in the readers.

How were the different rainfall intensity chosen? It is briefly said in the initial part of the

paper that this was based upon the rainfall values in South Korea, but then no rainfall data was provided to justify the choice of the adopted values. It would be good to add a few lines, or a figure, to describe the rainfall trend in the area. Further, a brief text explaining the importance of establishing relationships between rainfall and geological hazards could be useful, also referring to other hazards such as landslides (see for instance the works by Peruccacci et al. (2012), Rossi et al. (2012), and Vessia et al. (2012).

When quoting figures throughout the manuscript, please avoid he use of multiple brackets.

In general, the reference list can be improved, especially by adding the main international works about sinkhole classification, which are lacking in the present version of the manuscript. Apart from some references directly suggested in the accompanying file, I am enclosing to this comment a list of possible additional references that might be useful to the Authors to improve their paper.

When quoting more than one paper in the text, the references must be listed in chronological order. This guideline is not followed in the manuscript. Please correct it throughout the text.

Suggested references:

Beck, B.: Soil Piping and Sinkhole Failures. In: Encyclopedia of Caves (Second Edition), White, W. B. and Culver, D. C. (Eds.), Academic Press, Amsterdam, 2012. Closson D, Abou Karaki N (2009) Human-induced geological hazards along the Dead Sea coast. Environ Geol 58:371–380. Gutiérrez, F., Guerrero, J., Lucha, P., 2008. A genetic classification of sinkholes illustrated from evaporite paleokarst exposures in Spain. Environ. Geol. 53, 993–1006. Gutierrez F., Parise M., De Waele J. & Jourde H., 2014, A review on natural and human-induced geohazards and impacts in karst. Earth Science Reviews, vol. 138, p. 61-88, doi: 10.1016/j.earscirev.2014.08.002. Parise M., 2015, A procedure for evaluating the susceptibility to natural and anthropogenic sinkholes.

Georisk, vol. 9 (4), p. 272-285, DOI:10.1080/17499518.2015.1045002. Parise M., 2019, Sinkholes. In: White W.B., Culver D.C. & Pipan T. (Eds.), Encyclopedia of Caves. Academic Press, Elsevier, 3rd edition, ISBN 978-0-12-814124-3, p. 934-942. Parise M., Pisano L. & Vennari C., 2018, Sinkhole clusters after heavy rainstorms. Journal of Cave and Karst Studies, vol. 80 (1), p. 28-38. DOI: 10.4311/2017ES0105. Peruccacci, S., Brunetti, M. T., Luciani, S., Vennari, C., and Guzzetti, F.: Lithological and seasonal control on rainfall thresholds for the possible initiation of landslides in central Italy, Geomorphology, 139–140, 79–90, 2012. Rossi, M., Peruccacci, S., Brunetti, M. T., Marchesini, I., Luciani, S., Ardizzone, F., Balducci, V., Bianchi, C., Cardinali, M., Fiorucci, F., Mondini, A. C., Reichenbach, P., Salvati, P., Santangelo, M., Bartolini, D., Gariano, S. L., Palladino, M., Vessia, G., Viero, A., Antronico, L., Borselli, L., Deganutti, A. M., Iovine, G., Luino, F., Parise, M., Polemio, M., and Guzzetti, F.: SANF: a national warning system for rainfall-induced landslides in Italy, in: Proceedings of the 11th International Conference and 2nd North American symposium on landslides, Banff, Alberta, Canada, 3–8 June, 2012. Vessia G., Parise M., Brunetti M.T., Peruccacci S., Rossi M., Vennari C. & Guzzetti F., 2014, Automated reconstruction of rainfall events responsible for shallow landslides. Natural Hazards and Earth System Sciences, vol. 14, p. 2399-2408. Waltham, T., Bell, F., Culshaw, M., 2005. Sinkholes and Subsidence. Springer, Chichester, (382 pp.). White, W.B., 2002. Karst hydrology: recent developments and open questions. Eng. Geol. 65, 85–105.

For all the considerations above, I recommend minor revision. I believe that, after some corrections, and following the journal guidelines for citations, the manuscript may become acceptable for publication.

Please also note the supplement to this comment:
https://nhess.copernicus.org/preprints/nhess-2020-143/nhess-2020-143-RC2-supplement.pdf
* * *
2020-143, 2020.

**Supplement:**

[revised manuscript text omitted]

---

## Author Comment (AC1) · 17 Sep 2020

1. General comments to authors This paper examined the effect of rainfall intensity on the sewer-related soil erosion and its evolution by means of model tests and image analysis. In order to reflect the field conditions in South Korea, the backfill material, rainfall intensity, and compaction criteria were considered in the model tests. The topic is clear and suitable with the subject of this journal. There are, however, several aspects that need to be improved, especially in relation with the test procedure and actual sewer-related soil erosion. Revising the manuscript considering the following comments are also recommended.

Answer: Thank you very much for your thorough and helpful review. Based on your concerns and comments, we believe that our manuscript has been improved. Please check our answers corresponding to your concerns.

2. Specific comments to authors Q1: The terms 'ground cave-in' and 'sinkhole' are used interchangeably which are recommended to be unified.

Answer: Based on your comment, the authors changed the term "ground cave-in" to "sinkhole."

Q2: L101. "The width of the slit was set to 2 cm, based on the study by Mukunoki et al. (2012), such that B/Dmax was 4.2." Justify the width of the slit the authors determined in relation with the listed reference.

Answer: Mukunoki et al. (2012) adjusted the ratio B/Dmax between the slit width B and maximum grain size Dmax of the soil to 1.05, 2.5, and 5.9 in their model tests. A ground cavity was formed after 13 cycles when B/Dmax = 1.05, whereas a sinkhole was observed when both B/Dmax = 2.5 and 5.9. In the model tests that were performed in our study for the adjusted Gwanak soil, which describes the typical backfill materials for the underground pipes used in South Korea, B/Dmax = 4.2. (For the adjusted Gwanak soil, Dmax = 4.75 mm and B = 20 mm.) This B/Dmax value is between 2.5 and 5.9, corresponding to the sinkhole development described by Mukunoki et al. (2012).

Q3: L107-108. Some clarification on the condition of "Relationship between the rainfall intensity and the hydraulic head in the sewage network conditions near Gangnam station" are needed. I wonder if this condition has been sufficiently considered in the model tests of this study.

Answer: The test conducted by National Disaster Management Institute of Korea (2014) shows the relationship between the rainfall intensity and hydraulic head under the sewage network conditions near Gangnam station. The sewage network was simulated by considering the distance between each sewer pipes and burial depth of the

sewer pipes. In addition, the sewer pipes were assumed to be 1000 mm in diameter, as reflected in the present study. The authors noted these points in the manuscript.

Q4: L117. typo (#No. 4 sieve passing).

Answer: Based on your comment, the authors removed the term of "#No. 4 sieve passing."

Q5: L141-142. Additional information on the validation of PIV technique, such as accuracy, analysis condition, will be of interest to the readers.

Answer: In this study, to find the optimum size of the pixel subset, various-sized pixel subsets (40 × 40, 60 × 60, 80 × 80, 100 × 100, and 120 × 120) were tested by comparing two digital images: the original image of the model ground and the image artificially shifted by 10 pixels at the 4 edges of the model ground (where crude distortion occurs). The validation test results showed that 100 × 100 was the optimum size of the pixel subset, with a maximum error of 0.0069 pixels in accuracy and precision. The PIV validation results have been included in the manuscript.

Q6: L149. Explanation about the multiple cycles is required. I believe that one cycle consisted of water supply and drainage stage and it was repeated, but the manuscript contains only the result of one cycle.

Answer: Based on your comment, the authors removed the term "multiple cycles."

Q7: Additional information on the rainfall record which can prove the suggested three rainfall intensities in this study are realistic will enhance the credit of this paper.

Answer: Currently, the standard for a heavy rain watch in South Korea is 60 mm/3 h (in the case of intense heavy rain) or 110 mm/12 h (in the case of continuous heavy rain), and the standard for a heavy rain warning is 90 mm/3 h (in the case of intense heavy rain) or 110 mm/6 h (in the case of continuous heavy rain). Because the focus of this study was the formation of anthropogenic sinkholes in the event of intense heavy rainfall, the hourly rainfall intensity distributions corresponding to 60 mm/3 h and 90

mm/3 h (which are the criteria for a heavy rain watch and heavy rain warning) were confirmed using data from the Environmental Prediction Research Institute (2017) (as of 2012–2016). In the cases of 60 mm/3 h (based on a heavy rain watch) and 90 mm/3 h (based on the heavy rain warning level), the hourly rainfall intensity distributions corresponding to 30–50 mm/h and 40–60 mm/h were the highest, respectively. In the heavy rain watch case, the rainfall intensity distribution of 30–50 mm/h was 72.9 %, and in the heavy rain warning case, the rainfall intensity distribution of 40–60 mm/h was 64.9 %. Therefore, 40 mm/h and 50 mm/h were applied in this study by using the average value for the section with the highest rainfall distribution for 1 h in terms of heavy rain watch and heavy rain warning.

Q8: The procedure of calculating the average cavity width in Table 4 is not clear.

Answer: Based on your comment, the authors specified the calculation procedure for the average cavity width in the footnote of Table 4.

---

## Author Comment (AC2) · 17 Sep 2020

An experimental study about the relationships between rainfall intensity and development of sinkholes caused by damaged sewer pipes in Korea is described in the paper. The topic is certainly of interest to NHESS, and the work contains interesting data and considerations. I have listed in the accompanying file a number of small corrections, and a few requests of clarification on some issues that are not clear to me.

Answer: First we are very grateful for your thorough and helpful review. Based on your concerns and comments, we believe that our manuscript has been improved. Please check our answers corresponding to your concerns.

[Figure]

I discourage throughout the manuscript the use of the term "ground cave-ins", since this is not used in the international literature, and may induce confusion and misunderstandings in the readers.

Answer: Based on your comment, the authors changed the term "ground cave-in" to "sinkhole."

How were the different rainfall intensity chosen? It is briefly said in the initial part of the paper that this was based upon the rainfall values in South Korea, but then no rainfall data was provided to justify the choice of the adopted values. It would be good to add a few lines, or a figure, to describe the rainfall trend in the area. Further, a brief text explaining the importance of establishing relationships between rainfall and geological hazards could be useful, also referring to other hazards such as landslides (see for instance the works by Peruccacci et al. (2012), Rossi et al. (2012), and Vessia et al.(2012)).

Answer: Currently, the standard for a heavy rain watch in South Korea is 60 mm/3 h (in the case of intense heavy rain) or 110 mm/12 h (in the case of continuous heavy rain), and the standard for a heavy rain warning is 90 mm/3 h (in the case of intense heavy rain) or 110 mm/6 h (in the case of continuous heavy rain). Because the focus of this study was the formation of anthropogenic sinkholes in the event of intense heavy rainfall, the hourly rainfall intensity distributions corresponding to 60 mm/3 h and 90 mm/3 h (which are the criteria for a heavy rain watch and heavy rain warning) were confirmed using data from the Environmental Prediction Research Institute (2017) (as of 2012–2016). In the 60 mm/3 h (based on heavy rain watch) and 90 mm/3 h (based on heavy rain warning) cases, the hourly rainfall intensity distributions corresponding to 30–50 mm/h and 40–60 mm/h were the highest, respectively. In the heavy rain watch case, the rainfall intensity distribution of 30–50 mm/h was 72.9 %, and in the heavy rain warning case, the rainfall intensity distribution of 40–60 mm/h was 64.9 %. Therefore, 40 mm/h and 50 mm/h were applied in this study by using the average value for the section with the highest rainfall distribution for 1 h in terms of heavy rain

watch and heavy rain warning. In addition, following your comment, we added a brief description of the importance of establishing the relationships between rainfall and geological hazards with suggested references.

When quoting figures throughout the manuscript, please avoid he use of multiple brackets.

Answer: Based on your comment, the authors removed all of the multiple brackets in manuscript.

In general, the reference list can be improved, especially by adding the main international works about sinkhole classification, which are lacking in the present version of the manuscript. Apart from some references directly suggested in the accompanying file, I am enclosing to this comment a list of possible additional references that might be useful to the Authors to improve their paper.

Answer: Based on your comment, the authors included the main international works about sinkhole classification (as recommended by reviewer) in the manuscript.

When quoting more than one paper in the text, the references must be listed in chronological order. This guideline is not followed in the manuscript. Please correct it throughout the text.

Answer: Based on your comment, the authors rearranged the references according to chronological order.

Suggested references: Beck, B.: Soil Piping and Sinkhole Failures. In: Encyclopedia of Caves (Second Edition), White, W. B. and Culver, D. C. (Eds.), Academic Press, Amsterdam, 2012. Closson D, Abou Karaki N (2009) Human-induced geological hazards along the Dead Sea coast. Environ Geol 58:371–380. Gutiérrez, F., Guerrero, J., Lucha, P., 2008. A genetic classification of sinkholes illustrated from evaporite paleokarst exposures in Spain. Environ. Geol. 53, 993–1006. Gutierrez F., Parise M., De Waele J. & Jourde H., 2014, A review on natural and human-induced

geohazards and impacts in karst. Earth Science Reviews, vol. 138, p. 61-88, doi: 10.1016/j.earscirev.2014.08.002. Parise M., 2015, A procedure for evaluating the susceptibility to natural and anthropogenic sinkholes. Georisk, vol. 9 (4), p. 272-285, DOI:10.1080/17499518.2015.1045002. Parise M., 2019, Sinkholes. In: White W.B., Culver D.C. & Pipan T. (Eds.), Encyclopedia of Caves. Academic Press, Elsevier, 3rd edition, ISBN 978-0-12-814124-3, p. 934-942. Parise M., Pisano L. & Vennari C., 2018, Sinkhole clusters after heavy rainstorms. Journal of Cave and Karst Studies, vol. 80 (1), p. 28-38. DOI: 10.4311/2017ES0105. Peruccacci, S., Brunetti, M. T., Luciani, S., Vennari, C., and Guzzetti, F.: Lithological and seasonal control on rainfall thresholds for the possible initiation of landslides in central Italy, Geomorphology, 139–140, 79–90, 2012. Rossi, M., Peruccacci, S., Brunetti, M. T., Marchesini, I., Luciani, S., Ardizzone, F., Balducci, V., Bianchi, C., Cardinali, M., Fiorucci, F., Mondini, A. C., Reichenbach, P., Salvati, P., Santangelo, M., Bartolini, D., Gariano, S. L., Palladino, M., Vessia, G., Viero, A., Antronico, L., Borselli, L., Deganutti, A. M., Iovine, G., Luino, F., Parise, M., Polemio, M., and Guzzetti, F.: SANF: a national warning system for rainfall-induced landslides in Italy, in: Proceedings of the 11th International Conference and 2nd North American symposium on landslides, Banff, Alberta, Canada, 3–8 June, 2012. Vessia G., Parise M., Brunetti M.T., Peruccacci S., Rossi M., Vennari C. & Guzzetti F., 2014, Automated reconstruction of rainfall events responsible for shallow landslides. Natural Hazards and Earth System Sciences, vol. 14, p. 2399- 2408. Waltham, T., Bell, F., Culshaw, M., 2005. Sinkholes and Subsidence. Springer, Chichester, (382 pp.). White, W.B., 2002. Karst hydrology: recent developments and open questions. Eng. Geol. 65, 85–105. For all the considerations above, I recommend minor revision. I believe that, after some corrections, and following the journal guidelines for citations, the manuscript may become acceptable for publication.

Please also note the supplement to this comment: https://nhess.copernicus.org/preprints/nhess-2020-143/nhess-2020-143-RC2-supplement.pdf

[Figure]

Answer: The authors considered the comment made by the reviewer in the revision.
* * *

---

## Author Comment (AC3) · 17 Sep 2020

**1. General comments**

This paper reports a series of experiments to analyze the sinkhole formation associated with rainfall intensity by simulating a leakage of an underground damaged sewer pipe. A slit at the bottom of the experiment chamber was considered as damage of pipe, and three different rainfall intensities were designed by controlling the hydraulic head connected to the slit of the chamber. The ground settlement was measured, and the deformation of soils around the pipe is captured by the particle image velocimetry (PIV) technique. Overall, the authors in this work present a rising issue of the sinkhole and

its relationship to rainfall by utilizing an experimental model set-up. I believe this paper will be of interest to the audience and would support publication after the following comments are addressed.

Answer: Thank you very much for your thorough and helpful review. Based on your concerns and comments, we believe that our manuscript has been improved. Please check our answers corresponding to your concerns.

1) Authors should clearly address and explain how the test procedure is designed to simulate the rainfall and the sewer pipes. a. In the test procedure, the hydraulic head was selected as a variable to represent the rainfall intensity, which eventually formed different target groundwater levels. Therefore, the amount of water introduced into the chamber and the duration of water supply stage may indicate additional information related to the rainfall intensity. For example, if the water supply stage of Test 3 took longer than that of Test 1, this set-up may not properly reflect the actual rainfall intensity and its influence on the groundwater level around the pipes. In addition, the flow from the damaged sewer pipe may not be the only source of water supply into the underground.

Answer: The ground condition of each test is the same; thus, the amount of water introduced into the chamber which has a linear relationship with the groundwater level is also a function of the rainfall intensity. In addition, the time to form the target groundwater levels during the water supply stage was almost the same. In this study, we investigated the soil erosion due to damaged sewer pipes in urban areas by performing model tests. The large area of the ground surface in an urban areas is covered with impervious pavement. Thus, we presumed the flow from damaged sewer pipes to be the main source of water supply into the underground.

b. In this work, the damage of sewer pipes is approximated as a slit of which the size is determined followed by the previous study. While this damage could be the main source of water supply into the ground, the drainage may not occur through this damage. In other words, the water drainage set-up using the slit and drainage valve

may lead to an extra discharge of soils and water. If the groundwater is discharged through a thin slit at the bottom of the chamber, it may be easily expected that the soil around the slit may easily collapse and deformed.

Answer: In this study, the model test procedure consisted of (1) the water supply stage, in which sewer water infiltrates from the pipes to the ground through damaged sections during heavy rainfall periods, and (2) the water drainage stage, which describes the drainage of groundwater into the sewer pipes through the damaged sections after heavy rainfall periods. As the reviewer noted, overflows may occur in the other direction rather than through damaged sections. After heavy rainfall, however, the hydraulic pressure of a sewer pipe becomes lower (as the sewer pipe becomes vacant); thus, it is the most likely that the groundwater will flow back through the damaged portions. The authors found that a description of what each step actually meant in field conditions was lacking. Thus, appropriate descriptions have been added in the manuscript.

2. Specific comments

1) Line 149: What's the meaning of multiple cycles?

Answer: Based on your comment, the authors removed the term "multiple cycles."

2) Line 173: As point out by the authors, the test-set up is likely to the piping simulation. Then, is it possible to analyze the sinkholes and rainfall intensity though the piping analysis? For example, using the critical hydraulic gradient?

Answer: The authors think that the piping analysis is valid only for the water supply stage. During the water supply stage (which describes the situation during rainfall), the effective stress decreases because upward seepage occurs through the slit; therefore, piping could occur. During the water supply stage, the water pressure (as compared to the soil pressure) was not sufficient to induce the piping in test 1 and 2; however, in test 3, piping occurred during the water supply stage, which implies that the water pressure has sufficient magnitude to reduce the soil effective stress to zero. During the

water drainage stage, however, the use of the piping concept for sinkhole analysis is limited. The critical hydraulic gradient is usually valid when the seepage has a direction different from that of the gravitational force; however, the seepage and gravitational force have the same direction near the slit during the water drainage through the slit. In this case, soil particles freely fall with water and the ground cavity (or cave-in) expands until (apparent) cohesion recovers because of partial saturation. For all tests described in the manuscript, ground cavities (or cave-ins) occurred.

3) Line 255: How is the resistance factor determined? If there's an equation, it could be helpful for readers. 4) Line 262: Is the matric suction analyze in a quantitative manner? Because of cohesion, this may limitedly affect the behavior of soils.

Answer (for both 3) and 4)): The authors think that the ability of the model tests to enable quantitative evaluation of the resistance factor or matric suction is limited. The authors are planning a numerical parametric study (such as coupled analysis between soil and groundwater) for this purpose.

5) Lines 279-280: The meaning is not clear.

Answer: The authors decided that Lines 279 and 280 were unnecessary and removed them from the manuscript.